# *Acanthamoeba castellanii* Genotype T4: Inhibition of Proteases Activity and Cytopathic Effect by Bovine Apo-Lactoferrin

**DOI:** 10.3390/microorganisms11030708

**Published:** 2023-03-09

**Authors:** Gerardo Ramírez-Rico, Moises Martinez-Castillo, Roberto Cárdenas-Zúñiga, Daniel Coronado-Velázquez, Angélica Silva-Olivares, Mireya De la Garza, Mineko Shibayama, Jesús Serrano-Luna

**Affiliations:** 1Department of Cell Biology, Center for Research and Advanced Studies, Mexico City 07360, Mexico; gerardo.ramirez@cinvestav.mx (G.R.-R.); mireya.dela.garza@cinvestav.mx (M.D.l.G.); 2Faculty of Professional Studies, Autonomous National University of Mexico, Cuautitlan 54714, Mexico; 3Department of Experimental Therapeutics, The University of Texas MD Anderson Cancer Center, Houston, TX 77054, USA; mpcastillo@mdanderson.org (M.M.-C.); rcardenas@mdanderson.org (R.C.-Z.); 4Liver, Pancreas and Motility Laboratory, Unit of Research in Experimental Medicine, School of Medicine, Autonomous National University of Mexico, General Hospital of Mexico, Mexico City 06726, Mexico; 5Department of Infectomics and Molecular Pathogenesis, Center for Research and Advanced Studies, Mexico City 07360, Mexico; jdcoronado@cinvestav.mx (D.C.-V.);

**Keywords:** *Acanthamoeba castellanii*, bovine apo-lactoferrin, cytopathic effect, cysteine proteases, serine proteases

## Abstract

*Acanthamoeba castellanii* genotype T4 is a clinically significant free-living amoeba that causes granulomatous amoebic encephalitis and amoebic keratitis in human beings. During the initial stages of infection, trophozoites interact with various host immune responses, such as lactoferrin (Lf), in the corneal epithelium, nasal mucosa, and blood. Lf plays an important role in the elimination of pathogenic microorganisms, and evasion of the innate immune response is crucial in the colonization process. In this study, we describe the resistance of *A. castellanii* to the microbicidal effect of bovine apo-lactoferrin (apo-bLf) at different concentrations (25, 50, 100, and 500 µM). *Acanthamoeba castellanii* trophozoites incubated with apo-bLf at 500 µM for 12 h maintained 98% viability. Interestingly, despite this lack of effect on viability, our results showed that the apo-bLf inhibited the cytopathic effect of *A. castellanii* in MDCK cells culture, and analysis of amoebic proteases by zymography showed significant inhibition of cysteine and serine proteases by interaction with the apo-bLf. From these results, we conclude that bovine apo-Lf influences the activity of *A. castellanii* secretion proteases, which in turn decreases amoebic cytopathic activity.

## 1. Introduction

Free-living amoebae (FLA) are protozoa with a cosmopolitan distribution. This group of microorganisms can be isolated from soil, dust, air, drinking water, swimming pools, eyewash solutions, and contact lenses [1,2]. *Acanthamoeba* is a genus in the FLA group and is responsible for opportunistic and non-opportunistic infections in humans and other animals [2,3]. *Acanthamoeba castellanii* is a member of this genus, which promotes granulomatous amoebic encephalitis (GAE), cutaneous lesions, nasopharyngeal, and pulmonary infections in immunocompromised patients, whereas, in healthy people, it can cause amoebic keratitis (AK) [1]. During the infection process, these protozoa are in contact with immune factors, including the epithelial barrier, neutrophils, macrophages, mucus, immunoglobulins, and complement [4,5,6]. The mucus layer of mammals contains mucins, lysozyme, antimicrobial peptides, and lactoferrin (Lf), which are involved in the innate immune response [7]. Lf is an iron-binding glycoprotein that can exist in two functional forms: iron-associated (holo-Lf) and iron-free, apo-lactoferrin (apo-Lf). The latter is released by glandular epithelial cells and is found in the secondary granules of neutrophils [8,9]. The physiological concentration of this protein is in the range of 25 µM in mucosal secretions of the nose and eyes [10] and 80–90 µM in colostrum [11,12]. The increase in apo-Lf levels in tissues and the blood is usually associated with the inflammatory process induced by pathogens [8,13].

In 2012 it was demonstrated the protective effect of oral administration of Lf in the “dry eye syndrome” caused by an age-induced decrease in lacrimal gland secretory function [14]. It has also been well-documented that apo-Lf is able to protect the host in different ways during a microbial infection [15], including through bacteriostatic [16] and a bactericidal effects on Gram-negative and Gram-positive bacteria, causing permeability and damage in the outer membrane [17]. Additionally, Lf has been reported to have deleterious effects on the proliferation and colonization of a wide range of microorganisms, such as yeast, fungi, viruses, and protozoa [18,19,20,21,22,23]. Some studies on the interaction between FLA and apo-Lf have been reported; however, the microbicidal phenomenon was only partially evaluated, and the cellular mechanisms of apo-Lf activity on FLA are not completely understood [24]. Here, we studied the different microbiological effects of apo-Lf on *A. castellanii*. We found that *A. castellanii* possesses the ability to evade the amoebicidal and amoebostatic effects induced by bovine apo-Lf (apo-bLf). We also found that, despite this lack of effect, apo-bLf inhibits the cytopathic effect of *A. castellanii* on MDCK cells, possibly due to inhibition of the proteolytic activity of amoebic serine and cysteine proteases.

## 2. Materials and Methods

### 2.1. Culture of Acanthamoeba

*Acanthamoeba castellanii* genotype T4 (ATCC 50370) trophozoites were kindly provided by Dr. Simon Kilvington (Public Health Laboratory, Leicester, UK). These amoebae were obtained from human cases of *Acanthamoeba* Keratitis and were cultured at 25 °C in Chang’s medium supplemented with 10% (*v*/*v*) of fetal bovine serum (FBS; Equitech-bio, Kerrville, TX, USA). The identification and characterization were performed as previously described [25]. The *A. castellanii* were harvested during the logarithmic growth phase (48–72 h).

### 2.2. Viability Assays

To assess the viability of the amoebic cultures, trypan blue and FACs assays were performed. Briefly, 5 × 10^6^ trophozoites were incubated with different concentrations (25, 50, 100, 300, and 500 µM) of apo-bLf (NutriScience Innovations, LLC, Milford, CT, USA) for 1, 3, 6, and 12 h. The apo-bLf was diluted in the respective amoebic culture medium without FBS. Apo-bLf-free and serum-free cultures of *A. castellanii* were used as the experimental controls.

For the trypan blue technique, trophozoites were recovered after each apo-bLf treatment. The amoebae were then incubated with 0.5% trypan blue, and at least 100 cells from five different fields were analyzed by light microscopy (Primostar, Carl Zeiss, Madrid, Spain). The results are reported as the percentage of live amoebae. In addition, a quantitative analysis of viability was performed using a fluorescent SYTOX Green assay (Invitrogen, Waltham, MA, USA) according to the supplier’s protocol. Serum-free cultures and trophozoites fixed with 2% paraformaldehyde (Sigma Aldrich, St. Louis, MO, USA) were used as experimental controls. The data were obtained using a FACS Calibur flow cytometer (Becton Dickinson, San Jose, CA, USA) and analyzed using Summit 5.1 software (Becton Dickinson, San Jose, CA, USA). Three independent assays were performed for both viability assays. The statistical analysis was performed by one-way ANOVA using GraphPad Prism 5.0 (*p* > 0.05) (GraphPad, San Diego, CA, USA).

### 2.3. Growth Curves

*Acanthamoeba castellanii* cell division during the interaction with apo-bLf was also evaluated. Briefly, 5 × 10^6^ trophozoites were seeded in 15 mL sterile tubes (Corning, New York, NY, USA) and incubated with apo-bLf at the concentrations previously mentioned for 1, 3, 6, or 12 h. At each time interval, trophozoites were removed from the culture tube surface by chilling for 30 min. To determine the number of trophozoites, the samples were directly counted by light microscopy using a Bright-Line Hemacytometer (American Optical, Vernon Hills, IL, USA). As a control, the trophozoites were incubated in a serum-free medium. The statistical analysis of the three independent assays was performed by one-way ANOVA (*p* > 0.05) using GraphPad Prism 5.0 (GraphPad, CA, USA).

### 2.4. Transmission Electron Microscopy

For the ultrastructural analysis, the trophozoites were processed for transmission electron microscopy (TEM). The trophozoites were incubated with 500 µM of apo-bLf and fixed with 2.5% glutaraldehyde in 0.1 M of sodium cacodylate buffer (pH 7.2). Next, the samples were embedded in epoxy resin, and semi-thin sections were stained with toluidine blue (0.5%). The thin sections were contrasted with uranyl acetate and lead citrate and examined using an EM-910 Zeiss transmission electron microscope.

### 2.5. Cytopathic Effect of A. castellanii

The cytopathic effect on *A. castellanii* of different concentrations of apo-bLf was analyzed. Madin-Darby canine kidney (MDCK) cells were cultured in minimal essential medium (MEM) (Gibco, Grand Island, NY, USA) supplemented with 10% FBS in an atmosphere of 5% CO_2_ and at 37 °C for 48 h until they reached 95% confluence (monolayer). Subsequently, the trophozoites (1 × 10^6^) were co-cultured with MDCK cells in serum-free MEM at 37 °C in a 1:1 ratio in 24-well culture plates (Corning, Glendale, CA, USA) in the presence of 500-µM apo-bLf. After 12 h, the trophozoites were removed by chilling for 30 min, and the integrity of the cell monolayers was observed by light microscopy. The MDCK cells without trophozoites, the MDCK cells incubated with apo-bLf (500 µM), and the MDCK cells incubated with the trophozoites without apo-bLf were used as experimental controls. Three independent assays were performed under each condition.

### 2.6. Zymography Assays

The protease activity was determined by 10% SDS-PAGE co-polymerized with 0.1% porcine gelatin (Sigma Aldrich, St. Louis, MO, USA), bovine (NutriScience Innovations, LLC, CT, Milford, CT, USA), and human apo-lactoferrin (Sigma Aldrich, St Louis, MO, USA). Total crude extracts (TCEs) of trophozoites incubated with or without 50, 100, and 500 µM apo-bLf were obtained as previously described [26], with some modifications. Briefly, the trophozoites were removed from the culture flask surface by chilling in an ice bath for 30 min, centrifuged at 800× *g* for 10 min, and washed with PBS 1X (pH 7.2). Subsequently, the trophozoites were lysed in PBS 1X (pH 7.2) by six freeze–thaw cycles with hot water and liquid nitrogen. Conditioned medium (CM) was prepared according to a previously-described protocol [27]. Briefly, six million trophozoites were placed into culture flasks containing 3 mL of fresh serum-free Chang´s medium, then incubated at 37 °C for 12 h. The CM was removed and centrifuged at 1500× *g* for 10 min, and finally passed through a 0.22 µm Durapore membrane (Millipore, Bedford, MA, USA). Next, the samples were precipitated with absolute ethanol at a 3:1 ratio and stored at −20 °C for 2 h. The CM was centrifuged at 6000× *g* for 30 min. The protein concentrations of the TCEs and CM were quantified using the Bradford method [28]. Fourteen µg of protein were loaded onto a gel for SDS-PAGE, performed at 4 °C in an ice bath at 80 V for 3 h. The gels were then washed twice with 2.5% (*v*/*v*) Triton X-100 solution (Sigma Aldrich, St Louis, MO, USA) for 30 min and incubated overnight with 100 mM of Tris-HCl (pH 7.0) and 2 mM of CaCl_2_. Finally, the gels were stained overnight with 0.5% (*w*/*v*) Coomassie Brilliant Blue R-250 (Bio-Rad, Feldkirchen, Germany). The protease activity was identified as clear bands on the blue background. Three independent assays were performed for each sample.

### 2.7. Protease Inhibitors

For the proteinase inhibition assays, the TCEs and CM were preincubated for 1 h at 37 °C with different protease inhibitors under constant agitation. The concentrations of inhibitors were as follows: for the cysteine proteases, 10 mM of p-hydroxymercuribenzoate (pHMB), and 10 µM of E-64; for the serine and cysteine proteases, 5 mM of phenylmethylsulfonyl fluoride (PMSF), and 1 mM of aprotinin (all the inhibitors were from Sigma Aldrich, St Louis, MO, USA). We loaded the samples onto 10% SDS-PAGE co-polymerized with 0.1% bovine apo-lactoferrin (NutriScience Innovations, LLC, Milford, CT, USA), and then performed electrophoresis at 4 °C in an ice bath at 80 V for 3 h. The gels were then washed twice with 2.5% (*v*/*v*) Triton X-100 solution (Sigma Aldrich, St Louis, MO, USA) for 30 min and incubated overnight with 100 mM of Tris-HCl (pH 7.0) and 2 mM of CaCl2. The gels were then stained overnight with 0.5% (*w*/*v*) Coomassie Brilliant Blue R-250 (Bio-Rad, Feldkirchen, Germany). Three independent assays were performed for each sample (26).

## 3. Results

### 3.1. Acanthamoeba castellanii Resists the Microbicidal Effect of apo-bLf

To evaluate the effect of apo-bLf on *A. castellanii* viability, the trophozoites were incubated with different concentrations of the apo-bLf for different time periods. The Trypan blue assay showed 95% viability at 1, 3, 6, and 12 h at 500 µM (high concentration) of apo-bLf, as compared to that of the untreated trophozoites (Figure 1). We also observed that after 6 h of incubation with 500 µM of apo-bLf, there was an increase in the size of the trophozoites, and they were slightly rounded. 

Moreover, the cytometric analysis revealed that the *A. castellanii* trophozoites incubated with 500 µM of apo-bLf for 12 h maintained 98% viability (Figure 2c). These results were compared with those of the trophozoites cultured in the serum-free medium, which showed 97.2% viability (Figure 2a). As a positive control, the trophozoites were treated with paraformaldehyde for 1 h; the results showed about 7% trophozoite viability (Figure 2b). In both assays, we observed similar results; therefore, at physiological or even higher doses of apo-bLf (500 µM), there was no amoebicidal effect on the *A. castellanii* trophozoites.

### 3.2. Apo-bLf Does Not Have an Amoebostatic Effect on A. castellanii

It has been reported that apo-bLf induces amoebostatic effects; we therefore also evaluated the effect of apo-bLf on amoebic proliferation. *A. castellanii* trophozoites were incubated with different concentrations of apo-bLf, and the cultures were evaluated at 1, 3, 6, and 12 h. The counted cells revealed that the apo-bLf did not diminish the trophozoite growth rate, even at 500 µM apo-bLf (Figure 3). The growth rate at all time intervals was comparable to that in the serum-free medium (Figure 3); the differences were not statistically significant under any condition.

### 3.3. Ultrastructural Analysis of Trophozoites Treated with apo-bLf

To see whether there were morphological changes in amoebae treated with apo-bLf, we performed an ultrastructural analysis of *A. castellanii* trophozoites incubated with 500 µM of apo-bLf at the maximum time of study (12 h). The TEM analysis did not show any effect of the apo-bLf on the trophozoites. In the case of the apo-bLf-treated and non-treated *A. castellanii*, the amoebae showed typical morphological features of contractile vacuoles and nuclei (N) with a prominent nucleolus. The cytoplasmic membrane showed complete integrity, but with the presence of electron-dense material and an increase in the number and size of mitochondria (Figure 4).

### 3.4. Apo-bLf Inhibits the Cytopathic Effect of A. castellanii

To explore whether apo-bLf modifies the pathogenic capacity of *A. castellanii*, we performed in vitro assays using an epithelial cell line. The results showed that serum-free MDCK cells presented a normal morphology after 12 h, and similar results were observed in MDCK cells incubated only with apo-bLf (500 µM) (Figure 5a,b, respectively). In contrast, damage to the MDCK monolayer was evident in the presence of *A. castellanii* (Figure 5c). Interestingly, the cytopathic effect on the MDCK monolayers was reduced by the apo-bLf in co-cultures with *A. castellanii* after 12 h of incubation (Figure 5d).

### 3.5. Cysteine and Serine Proteases of A. castellanii Degrade Human and Bovine apo-Lactoferrin

To evaluate whether *A. castellanii* produces specific proteases with activity against apo-Lf, we performed zymography using human and bovine apo-Lf as substrates. Moreover, to evaluate the type of proteases involved in apo-bLf degradation, we used specific (E-64 and pHMB), partial (PMSF), and non-specific (aprotinin) inhibitors of cysteine proteases. The pHMB inhibitor completely decreased the proteases detected in TCEs (Figure 6a, lane 2), while the PMSF and E-64 slightly decreased the proteolytic profiles (Figure 6a, lanes 3 and 4). Additionally, the pHMB and PMSF inhibitors caused a reduction in the proteolytic patterns of the CM (Figure 6b, lanes 2 and 4). In addition, the results showed that the E-64 and aprotinin did not decrease the activity of the CM proteases (Figure 6b, lanes 3 and 5) compared to the CM not treated with the protease inhibitors. Similar results were obtained when we performed zymograms with apo-hLf, and two bands of proteolytic activity were found at 100 kDa and 55 kDa in the TCEs and CM, respectively (Appendix A).

### 3.6. Apo-bLf Inhibits the Proteolytic Profile of A. castellanii

*Acanthamoeba castellanii* trophozoites produce and release a broad spectrum of proteases that are involved in different cellular and pathogenic mechanisms. Therefore, we decided to evaluate the effect of apo-bLf on amoebic protease activity. We used 10% SDS-PAGE co-polymerized with 0.1% porcine gelatin as the substrate. When 50 µM of apo-bLf was incubated with *A. castellanii* trophozoites for 1 h, a decrease in the proteolytic activity of the TCEs was observed (Figure 7a,b, lanes 3–4); while, with 500 µM of apo-bLf, the activity almost disappeared, with only a 150 kDa protease at the evaluated times (Figure 7a,b, lane 4) compared with the serum-free trophozoites without apo-bLf (Figure 7a,b, lane 1). In contrast, when we analyzed the CM, the *A. castellanii* showed that the proteolytic pattern was completely inhibited in the presence of apo-bLf at 12 h (Figure 7c,d, lanes 2–4) compared to the CM obtained from the trophozoites incubated with the serum-free medium without apo-bLf (Figure 7c,d, lane 1). All the gels were activated at 37 °C and pH 7.0.

Interestingly, in the assays zymography co-polymerized with 0.1% apo-bLf, the TCEs of *A. castellanii* that were not incubated with apo-bLf showed bands of proteolytic activity at 130 and 87.5 kDa, (Figure 8a,b, lane 1). In contrast, pre-incubation of the trophozoites with different concentrations of apo-bLf showed that the 130 kDa activity was maintained, but the 87.5 kDa activity was inhibited after 6 h (Figure 8a,b, lanes 2–4). Moreover, after 12 h of incubation, no proteolytic activity was observed (Figure 8b). When we analyzed the CM from the *A. castellanii*, we found two proteolytic activities of 130 and 70 kDa (Figure 8c,d, lane 1); however, these activities were abolished after 1 h of incubation with apo-bLf (Figure 8c,d, lanes 2–4). 

## 4. Discussion

In mucosal tissues, several immune proteins, such as mucins, macroglobulins, immunoglobulins, defensins, lysozyme, and lactoferrin, actively participate in the elimination of pathogens [7]. Apo-lactoferrin is a multifunctional protein that is released by the mucoepithelial cells and secondary granules of neutrophils [8]. It has been implicated in several physiological functions, such as regulation of iron levels, anti-inflammatory activity, regulation of cellular growth and differentiation, protection against cancer, and host defense against a broad range of microbial infections [15,29,30,31]. It is important to mention that the microbicidal effect of apo-Lf in several microorganisms is not well understood; however, it seems to be related to bacterial membrane binding, which could alter membrane permeability through the dispersion of lipopolysaccharides and lead to cell death [17,29,31,32]. In contrast, apo-bLf is known to have fungicidal and antiparasitic activities [24,29,33,34].

During an *A. castellanii* infection of the eyes, these microorganisms interact directly with the apo-Lf from tears [35]. The interaction of apo-Lf with *A. castellanii* has been previously evaluated, showing an amoebicidal effect at a dose of 10 µM [36]. Initially, we wanted to know whether physiological concentrations of apo-bLf were able to kill *A. castellanii* trophozoites; however, we found that this protozoan was resistant. Therefore, we tested higher concentrations of apo-bLf, up to 500 µM; at all concentrations, we found viability greater than 95% of the trophozoites. We tested whether these high concentrations of apo-bLf could affect metabolic functions, such as virulence, of *A. castellanii*. Differences from previous studies could be due to the number of trophozoites that the authors reported (2 × 10^6^) compared with our study (5 × 10^6^). Another important difference may be related to the isolation method or strain of *A. castellanii* employed in each study (we employed *A. castellanii* ATCC 50370), which means that, in terms of specific *Acanthamoeba* strains, the susceptibility or resistance to Lf needs to be explored. In this context, a study of the incubation of *A. castellanii* with 1.48 mg/mL (19.24 µM) of Lf found that it does not cause alteration in the adhesion of *A. castellanii* on the SV40-human cornea epithelial cell line; however, the authors did not report the death of trophozoites under this condition [35].

Another mechanism related to apo-bLf is the microbiostatic effect, which has been correlated with the ability of Lf to chelate iron and render it unavailable to microorganisms [9,16,20]. This microbicidal effect has been reported in bacteria, fungi (*Candida* species), and protozoans such as *Toxoplasma gondii*, *Trypanosoma brucei*, *T. cruzi*, *Giardia lamblia* and *Babesia caballi* [18,19,37]. Interestingly, in our study, the growth curves did not show a microbiostatic effect of apo-bLf in *A. castellanii* cultures, even at higher concentrations and after 12 h of incubation. However, in the cultures of *A. castellanii* incubated with apo-bLf after 6 h, we observed some morphological changes by light microscopy, such as the amoebae becoming larger and slightly rounded. Moreover, in the TEM study, a thickening of the plasma membrane and an accumulation of electron-dense material in the cytoplasm were observed. We also observed an increase in the number of mitochondria, with an increase in their size with respect to the amoebae that were not treated with apo-bLf, which could be explained by a possible interaction between the apo-bLf and the *A. castellanii* trophozoites at the membrane level. The binding of Lf to the surface of *E. histolytica* trophozoites has also been reported [38]. A similar analysis was performed for pathogenic *Candida* species, where cell wall changes induced by apo-bLf were observed, causing eventual cell death [33]. However, in *A. castellanii* trophozoites, the antimicrobial activity of apo-bLf was not observed at higher concentrations. Even though there is no apparent effect on amoebic viability at high concentrations of apo-bLf, interestingly, when exploring the cytopathic effect of these amoebae on MDCK cells, inhibition of cell monolayer destruction was observed. This protective effect is consistent with a study reported by Coronado-Velazquez et al., where MDCK cells were incubated with *A. mauritaniensis* cells that were previously preincubated with serine protease inhibitors, resulting in an abolition of the cytopathic effect and a reduction in the degradation of epithelial tight junction proteins, demonstrating that *Acanthameoba* proteases are an important virulence factor for this pathogen [25]. The cytopathic effect is due to the secretion of cysteine and serine proteases. [25,26,27]. This led us to study the protease pattern of *A. castellanii* in both TCEs and CM. Protease inhibition revealed that cysteine proteases present in TCEs from *A. castellanii* trophozoites were involved in apo-bLf degradation. The proteases present in CM without apo-bLf revealed that cysteine and serine proteases could degrade the apo-bLf and holo-bLf. We previously reported that the secretion of cysteine proteases from *A. castellanii* is capable of degrading holo-Lf [27]. We found proteases with molecular weights of 56.6, 100, and 120 kDa in the TCEs and 60.2 and 100 kDa in the CM. They may be the same proteases that degrade apo-Lf, because the proteolytic degradation profiles were similar between both states of Lf, as well as non-specific, because these proteases are also capable of degrading other iron-associated proteins [27]. The degradation of holo-bLf by cysteine proteases present in the TCE of *E. histolytica* correlated with our results, where cysteine activities at 250, 100, 40, and 22 kDa cleaved holo-bLf at pH 7.0 and pH 4.0 [38]. Lf could bind to the cell membrane and be later internalized by the endosomal route to be degraded by cysteine proteases and used as a source of amino acids or as iron by cells, and apo-bLf could change to holo-bLf, due to iron chelation from the medium, and perhaps use the same pathway as reported in *E. histolytica* [38].

However, when *A. castellanii* trophozoites were incubated with apo-bLf, the protease secretion was inhibited and no proteolytic degradation was observed. Similar results were observed in zymograms co-polymerized with apo-bLf. These results could be due to apo-bLf functioning as a protease inhibitor or somehow inhibiting their secretion. These results are similar to those found in other microorganisms, such as *Actinobacillus pleuropneumoniae* and *Mannheimia haemolytica* [9,39]. In the enteropathogenic bacteria *Escherichia coli* (EPEC), which causes disease in several species, it has been reported that Lf damages the type III secretion system, the main secretion system of virulence factors in this bacterium, blocking actin polymerization and the degradation of EspB [40]. In our study, less inhibition was observed at lower concentrations of apo-bLf. Furthermore, in the COVID-19 pathology, caused by the viral agent SARS-CoV-2, proteolytic inhibition by peptides derived from bLf has been reported [41]. However, it is necessary to perform other assays, such as molecular docking, between proteases isolated from *A. castellanii* and apo-bLf and peptides derived from bLf (lactoferricin, lactoferrampin, and lactofungin).

## 5. Conclusions

Our results demonstrated that *A. castellanii* is one of the few microorganisms that can resist the amoebecidal or amoebestatic effects of apo-bLf. However, the apo-bLf inhibited the protease activity of *A. castellanii* and drastically reduced the cytopathic effects of the trophozoites on MDCK cells. Therefore, we believe that amoebic proteases are a key virulence factor during pathogenesis, and apo-bLf could be considered as a possible therapeutic alternative against *A. castellanii*.

## Figures and Tables

**Figure 1 microorganisms-11-00708-f001:**
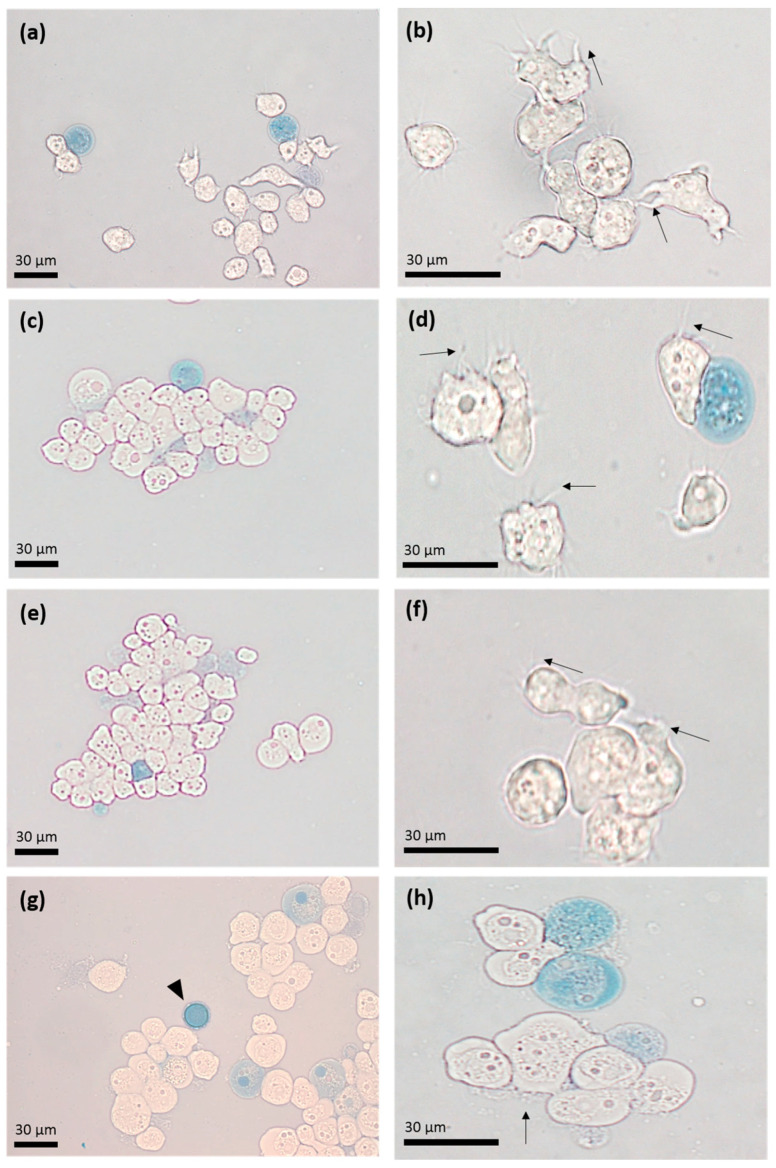
Light microscopy with Trypan blue exclusion assay. *Acanthamoeba castellanii* trophozoites were incubated with 500 µM of apo-bLf at 1 (**a**,**b**), 3 (**c**,**d**), 6 (**e**,**f**), and 12 (**g**,**h**) hours of incubation. Dead trophozoites (blue) are seen throughout the displayed fields. The presence of some trophozoites with acanthopods (arrows) and cysts (arrowheads) stands out. Representative results of at least three trials, each with independent samples.

**Figure 2 microorganisms-11-00708-f002:**
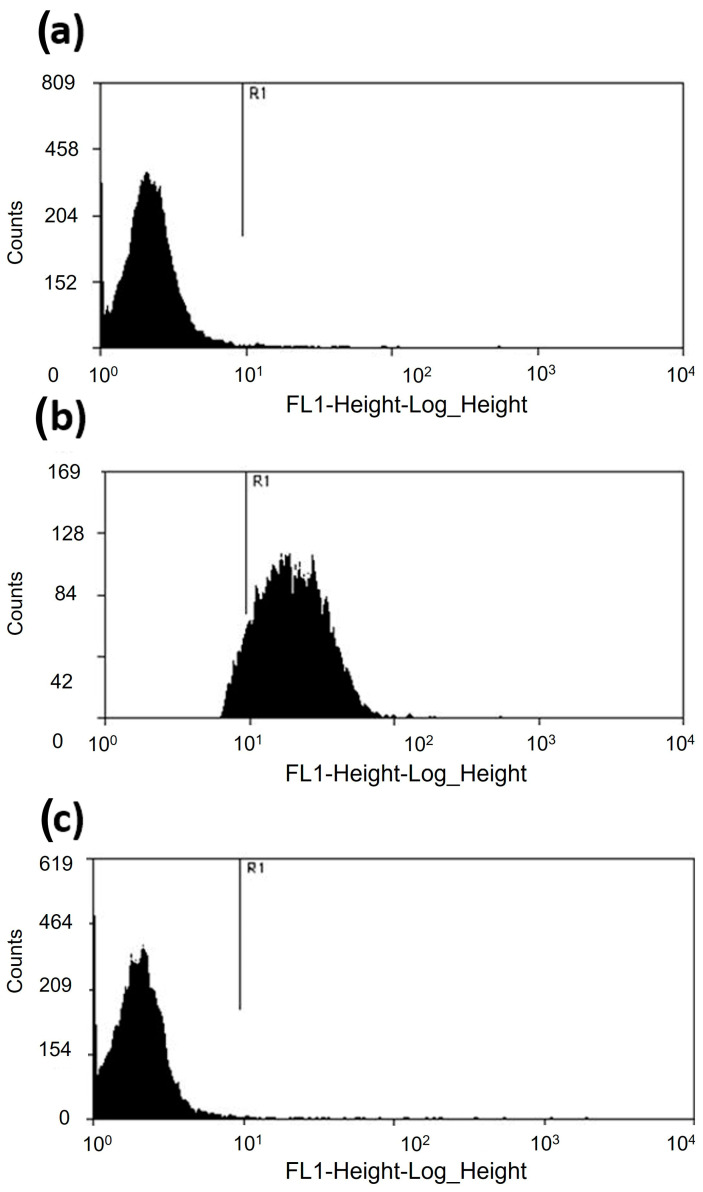
Viability of *A. castellanii* trophozoites incubated with 500 µM of apo-bLf. (**a**) About 97.2% of the *A. castellanii* trophozoites showed a weak SYTOX fluorescent marker at 12 h in the serum-free medium. (**b**) Trophozoites treated with paraformaldehyde for 1 h showed that the viability of *A. castellanii* was reduced to 7%. *Acanthamoeba castellanii* co-incubated with 500 µM of apo-bLf showed 98% viable cells. (**c**) The dead trophozoites were labeled with SYTOX Green (15 nM) and evaluated in a FACSCalibur flow cytometer. The results were analyzed with Summit 5.1 software. At least 20,000 events were counted for each condition. The results are expressed as percent of viable cells. Three independent assays were performed (*p* > 0.05).

**Figure 3 microorganisms-11-00708-f003:**
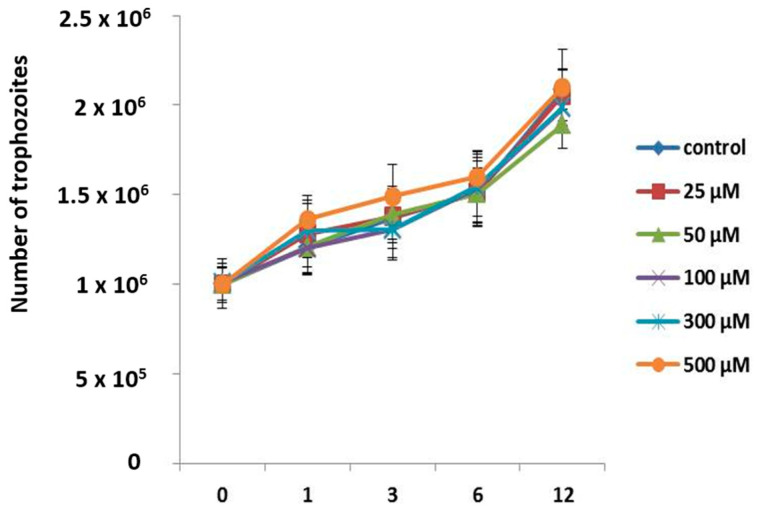
Growth curves of *A. castellanii* trophozoites incubated with different apo-bLf concentrations. Cellular division of *A. castellanii* was maintained at all apo-bLf, concentrations tested (25, 50, 100, 300, and 500 µM). Trophozoites in the serum-free medium were used as the experimental control (Chang). Data analyses were performed with GraphPad Prism 5.0. All bars show the mean ± SEM of three independent assays (*p* > 0.05).

**Figure 4 microorganisms-11-00708-f004:**
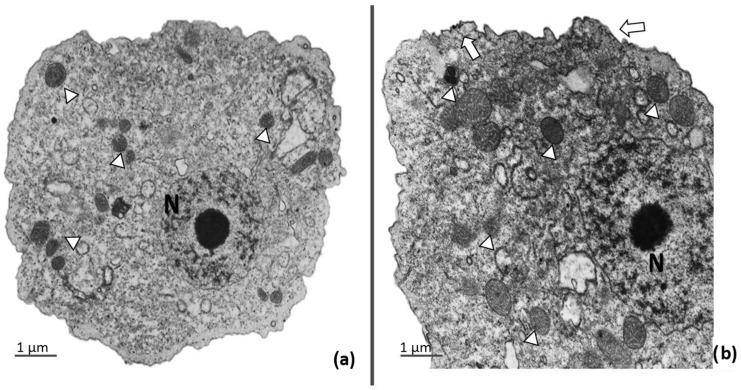
Transmission electron microscopy of *A. castellanii* after apo-bLf incubation. (**a**) *A. castellanii* trophozoites were incubated in serum-free medium for 12 h. They present a normal nucleus (N) with prominent nucleolus, intact membrane, and multiple healthy mitochondria; (**b**) *A. castellanii* incubated with 500 µM of apo-bLf. Some trophozoites showed high density in the cytoplasmic membrane (arrows) and an increase in intracytoplasmic electron-dense material, and high density in the mitochondria (arrowheads). Both, *A. castellanii* untreated and treated with 500 µM of apo-bLf show integrity of organelles and cytoplasmic membrane. EM-910 Zeiss. Bars = 1 µm.

**Figure 5 microorganisms-11-00708-f005:**
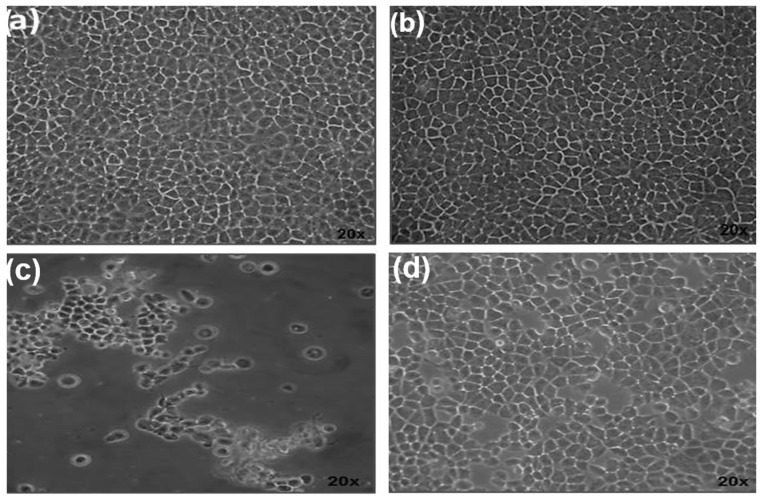
Inhibition of cytopathic effect by apo-bLf in *A. castellanii*. (**a**) MDCK monolayer with serum-free medium and (**b**) with apo-bLf (500 µM) showed a continuous typical monolayer morphology at 12 h of culture. (**c**) Alteration of MDCK cells by *A. castellanii* at 12 h. (**d**) Inhibition of cytopathic effect of *A. castellanii* by apo-bLf at 12 h of co-incubation.

**Figure 6 microorganisms-11-00708-f006:**
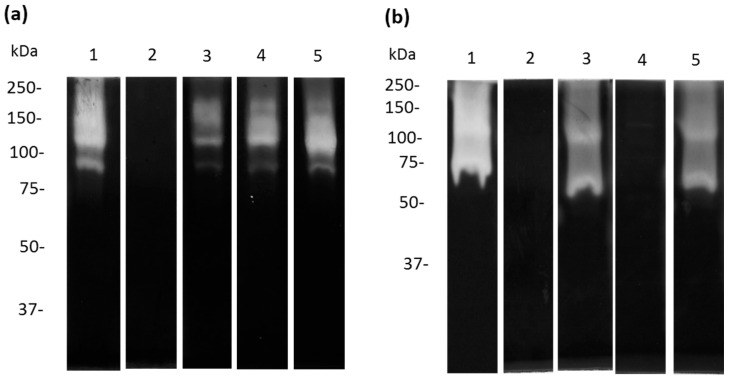
Inhibition of apo-bLf proteases from TCE and CM from *A. castellanii*. Zymography using 10% SDS-PAGE co-polymerized with 0.1% apo-bLf. Samples of TCE (**a**) and CM (**b**) from *A. castellanii* trophozoites were incubated for 1 h with different protease inhibitors: pHMB (lane 2), E-64 (lane 3), PMSF (lane 4), and aprotinin (lane 5). Untreated *A. castellanii* trophozoites without inhibitors were used as the experimental control ((**a**,**b**) lanes 1). The gels were activated overnight at 37 °C and pH 7.0. Three independent assays were performed.

**Figure 7 microorganisms-11-00708-f007:**
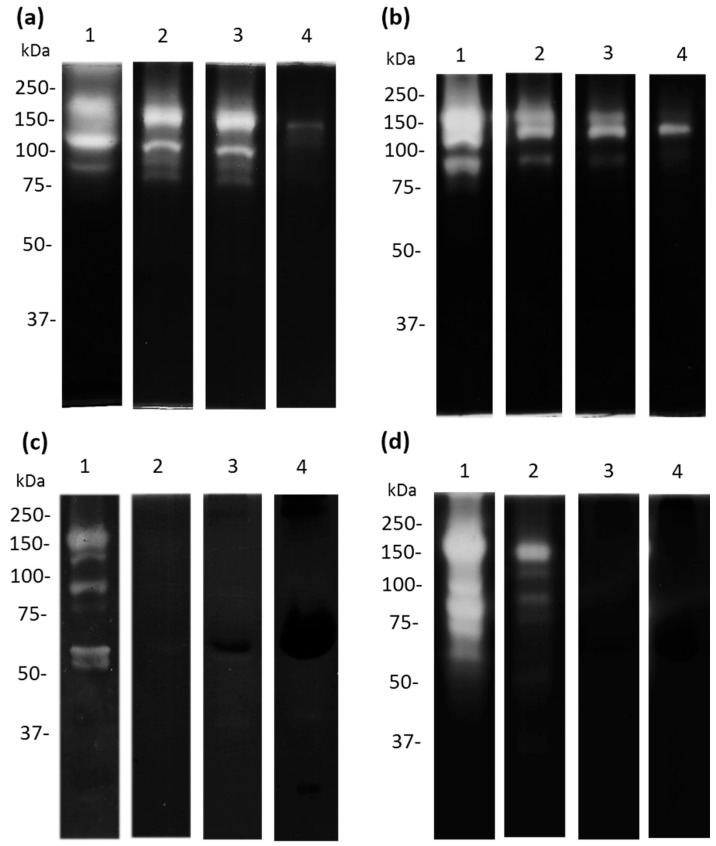
Proteolytic patterns from *A. castellanii* after apo-bLf incubation. Zymography using 10% SDS-PAGE co-polymerized with 0.1% porcine gelatin. Proteases of TCEs ((**a**,**b**) lane 1) and CM ((**c**,**d**) lane 1) from *A. castellanii* without incubation with apo-bLf. TCEs (**a**,**b**) and CM (**c**,**d**) from *A. castellanii* obtained after 1 h (**a**,**c**), and 12 h (**b**,**d**) of co-incubation with different concentrations of apo-bLf: 50 µM (Lane 2), 100 µM (lane 3), and 500 µM (lane 4). Three independent assays were performed.

**Figure 8 microorganisms-11-00708-f008:**
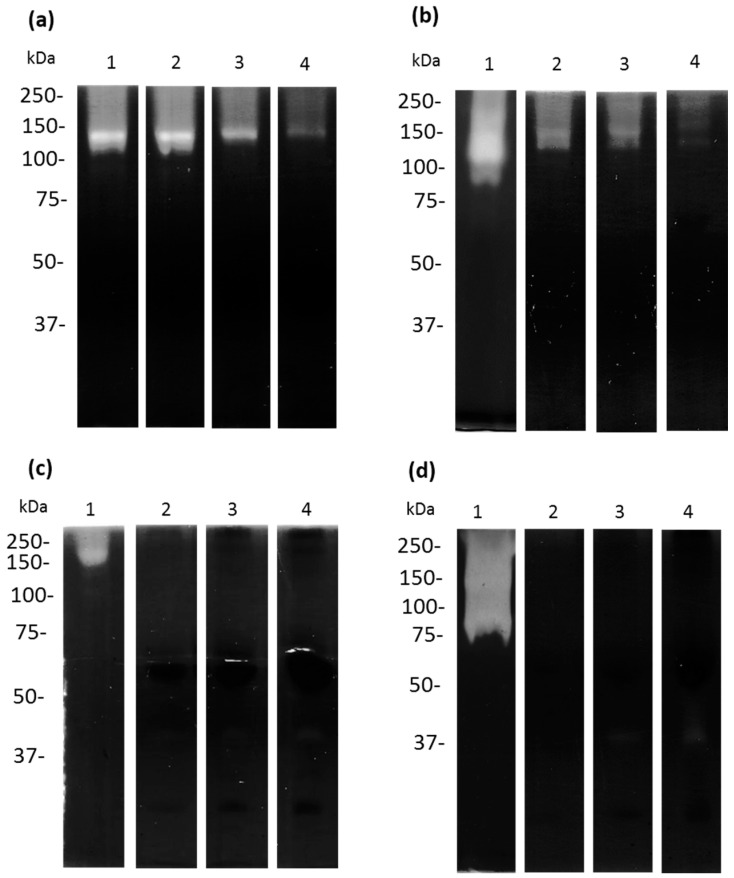
Proteases of TCEs and CM in zymograms using 10% SDS-PAGE co-polymerized with 0.1% bovine apo-lactoferrin. Untreated (lane 1) and treated-bLf (2–4) TCEs (**a**,**b**) and CM (**c**,**d**) from *A. castellanii* showed apo-bLf degradation. TCEs and CM from *A. castellanii* obtained at 1 h (**a**,**c**) and 12 h (**b**,**d**) of co-incubation with different concentrations of apo-bLf: 50 µM (lane 2), 100 µM (lane 3), and 500 µM (lane 4). All gels were activated at pH 7.0 and 37 °C. Three zymography assays with independent samples were performed.

## Data Availability

Data sharing not applicable. No new data were created or analyzed in this study. Data sharing is not applicable to this article.

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
