# Peer review of "Acanthamoeba castellanii Genotype T4: Inhibition of Proteases Activity and Cytopathic Effect by Bovine Apo-Lactoferrin"

_microorganisms, 2023, doi:10.3390/microorganisms11030708_

Round 1
Reviewer 1 Report
This manuscript finds for the first time that the mammalian defence protein lactoferrin inhibits the cytopathic effect of Acanthamoeba probably by its ability to inhibit the proteases expressed by the amoeba. This is an important discovery since Acanthamoeba causes a painful and dangerous disease called Acanthamoeba Keratitis. The paper is very well written and the conclusions justified by the results.
The title and the first line of the abstract focus specifically on Acanthamoeba castellanii but we know that AK is caused by Acanthamoeba other than those described as being “castellanii” and also that this classification is probably not valid (See Corsaro 2020).
It is a curious that the present authors saw no effect on cell viability caused by apo-Lactoferrin, while others (Tomita et al, 2017) have. One possible explanation is that the strains of Acanthamoeba used are behaving differently. The study that found reduced viability used a strain named Acanthamoeba sp. AA014 from an AK case, and the present study used a “Acanthamoeba castellanii” strain from an AK case given to them by Dr. Simon Kilvington. In light of this it is important to get as much information about this strain as possible. Are there any other details of this strain available? Has this strain been found to belong to the T4 genotype? What is the basis for it being described as Acanthamoeba castellanii?
Were any other Acanthamoeba strains tested in this study?
Given that the major conclusion of the study is that Acanthamoeba is able to resist the cytocidal effect of apo-bLf it would be beneficial to be able to demonstrate that this particular source of apo-bLf actually has the expected cytocidal effect on other microorganisms. Ideally the Acanthamoeba strain AA014 should be tested also, but this may not be available?
The majority of these experiments were conducted using bovine lactoferrin, but some used human lactoferrin. It would be important to know if the human lactoferrin was able to protect MDCK monolayers from Acanthamoeba as the bovine source did? The same is true for other experiments. I suppose this will be investigated in other studies from this group?
Also probably for a further study. It has been found that not only does lactoferrin block Acanthamoeba proteases but these proteases also digest lactoferrin. What is the nature of this lactoferrin proteolysis, what are the products and do these digested lactoferrin polypeptides retain their ability to inhibit Acanthamoeba proteases?
It is important to mention that others have found lactoferrin to inhibit various proteases (Ochoa et al, 2003; Zhao et al, 2022)
Corsaro, D. (2020). Update on Acanthamoeba phylogeny. Parasitology Research, 119(10), 3327-3338.
Ochoa, T. J., Noguera-Obenza, M., Ebel, F., Guzman, C. A., Gomez, H. F., & Cleary, T. G. (2003). Lactoferrin impairs type III secretory system function in enteropathogenic Escherichia coli. Infection and immunity, 71(9), 5149-5155.
Tomita, S., Suzuki, C., Wada, H., Nomachi, M., Imayasu, M. & Araki-Sasaki, K. (2017) Effects of lactoferrin on the viability and the encystment of Acanthamoeba trophozoites. Biochem Cell Biol 95, 48-52, doi:10.1139/bcb-2016-0054.
Zhao, W., Li, X., Yu, Z., Wu, S., Ding, L., & Liu, J. (2022). Identification of lactoferrin-derived peptides as potential inhibitors against the main protease of SARS-CoV-2. LWT, 154, 112684.
Author Response
I very much appreciate all the comments and observations made to the manuscript.
In a file attached you will see all the answers to your questions. I hope you find them satisfactory.

Reviewer 2 Report
This is an interesting study, generally well-written and overall clearly presented. However, while the lactoferrin results are convincing, and it is really interesting that Acanthamoeba seems to be resistant, the protease assays are not really convincing to me. Possibly also because some parts of the methodological set-up are unclear.
Specific comments:
lines 70-76: this is not clear to me, which strain(s) was/were used? Strain name? Is this strain available in any collection, e.g. ATCC or CCAP?
In line 72, the authors state that the amoebas were obtained from human cases, so it must be more than one strain!?
line 79 (and lines: 97, 114 etc.): I guess the "6" has to be put up (as the exponent)
lines 113-120: also this part is unclear to me, were the trophozoites and MDCK cells really inoculated at the same time into the wells? Usually, the MDCK cells are first grown to a monolayer and then (24-48 hours later) the trophozoites are added. And at wich temperature? If room temperature (as described for the amoebas (reference 25), this will not allow the MDCK cells to form a monolayer.
lines 122-142: this has to be described in more detail, e.g. how/why were the gels sliced up? what was incubated separatedly with what?
line 132: why were the cells here incubated at 37°C, while for all other experiments, even the cocultures at room temp?
line 150: the authors write "as previously described", but there is no reference for this. Also, this section has to be described in more detail.
Figure 1: what does the scale bar mean? Is it really µM (micromolar?), and if it is µm, this cannot be correct, as cells in 1g are double the size as in e.g. in 1e; as same scale bar. Please check and correct.
Figures 6-8: these zymograms look very strange, proteins did not separate well, it rather looks as if something went wrong during zymography. This part is not convincing.
e.g. Fig 6, lane 1 is a smear, particularly in 6b; typically in these zymograms, TCEs have a clear banding pattern with numerous bands; CMs less pronounced, but still visible.
line 293: I suggest to write: cell death (instead of: death of the organism)
line 338: why is the secretion of proteases due to proteases? I think the authors mean that the cytopathic effect is due to secretion of cysteine and serine proteases.
line 371: the authors state: "could be considered as a possible therapeutic alternative or as a complementary option", but how? this has not been mentioned or discussed in the Discussion section. What would be possible therapeutic options? and for which infection (eye/systemic)? What is the evidence for this?
Author Response
I very much appreciate all the comments and observations made to the manuscript.
I hope that the responses to your comments are satisfactory.

Round 2
Reviewer 2 Report
The authors have addressed all points raised. They have corrected all errors and better explained everything, but I am still NOT convinced at all by the zymograms. With my previous review I actually tried to encourage the authors to repeat them. Otherwise, I would suggest to exclude them.
I cannot see the marker in the figure provided.
Also, the scale bars in figure 1 still look rather arbitrary to me.
Author Response
Reviewer 2, Round 2
We greatly appreciate the revision and the time invested in our manuscript, your criticisms and clarifications undoubtedly improve the quality of the manuscript (MS).
Comments and Suggestions for Authors
The authors have addressed all points raised. They have corrected all errors and better explained everything, but I am still NOT convinced at all by the zymograms. With my previous review I actually tried to encourage the authors to repeat them. Otherwise, I would suggest to exclude them.
I cannot see the marker in the figure provided.
Figures 6-8: these zymograms look very strange, proteins did not separate well, it rather looks as if something went wrong during zymography. This part is not convincing.
e.g. Fig 6, lane 1 is a smear, particularly in 6b; typically in these zymograms, TCEs have a clear banding pattern with numerous bands; CMs less pronounced, butstill visible.
Answer: Thank you for your valuable comments.
We want to mention that the zymography assay is a very well standardized technic in our laboratory. However, we would like to emphasize that the electrophoresis that we are presenting in figures 6, 7 and 8 are complete. We know that the proteolytic bands that appear in the gels depend on the substrate we used, so we have that the proteases that appear in a gelatin gel are different from those of apo-bLf. As example, we present an article published in 2015 by our group (Ramírez-Rico et al, 2015). We reported the presence of different proteolytic activities in both TCE and CM of Acanthamoeba castellanii depending on the copolymerized substrate (human holo-Lf, human holo-Tf, human Hemoglobin and equine Ferritin). In the Figure 2A we showed the proteolytic activities corresponding to a human holo-Lf, TCE (lanes 1, 2, and 3) and CM (lanes 4, 5, and 6) of Acanthamoeba castellanii present a proteolytic pattern that is similar to that present in the Figure 8 of the present manuscript. We also want to add some replicates of zymography assays to show you the reproducibility of the experiments of the present study, including the molecular weight markers, which supports that the resolution of electrophoresis was complete. In the Figure A, we show TCE of A. castellani incubated with different concentrations of apo-bLf; lane 1 (Molecular weight markers), lane 2, 3 and 4 (50, 100 and 500 µM of apo-bLf respectively), lane 5 (TCE without apo-bLf). Figure B, we show CM of A. castellanii incubated with different concentrations of apo-bLf , lane 1 (Molecular weight markers), lane 2 (100 µM of apo-bLf), lane 3 (CM without apo-bLf).
We consider that the data provided in this manuscript are novel and valuable information. In addition, we also want to emphasize the importance of showing in the gels the inhibition of proteolytic activities by apo-bLf and thus suggest that the inhibition of the cytopathic effect of A. castellanii trophozoites could be due to the inhibition of the proteolytic activities by apo-bLf.
Nevertheless, we will consider the comment of the reviewer for future studies focusing in the purification and characterization of each activity present in the zymograms.
On the other hand, several papers published by different authors showed different profile of proteases in Acanthamoeba sp. that varies according to the authors. It can also be seen that in some gels a smear of the proteases is shown.
In the report from round 1, the reviewer mentioned:
Figure 1: what does the scale bar mean? Is it really µM (micromolar?), and if it is µm, this cannot be correct, as cells in 1g are double the size as in e.g. in 1e; as same scale bar. Please check and correct. Additionally, in the round 2, the reviewer writes: Also, the scale bars in figure 1 still look rather arbitrary to me.
Answer: We want to mention that the mistake of µM (micromolar) was corrected and we add µm following the reviewer observation.
About the size of the bar, it is important to mention that the scale was automatically burned by the software Nis Elements-Br from the Nikon Eclipse 80i microscope, thus is not an arbitrary size.
Regarding of the size of the trophozoites, we follow this observation in the Result section since the first version of the MS. In the line 174-176 we wrote the statement: “We also observed that after 6 h of incubation with 500 μM apo-bLf, there was an increase in the size of the trophozoites, and they were slightly rounded”, moreover this observation was also included in the lines 363-364 in the Discussion section. In other words, the size of the trophozoites increased and the scale bar was just used to confirm that point.
It is important to clarify that these morphological changes were mainly observed at 500 uM, which is a non-physiological concentration of apo-Lf and any variation in morphology induced by this concentration could be considered as a mechanism related with volume homeostasis (hypertonic/isotonic feature) that should be considered important to elucidated in further studies.